# Mitochondrial Variation of Bottlenose Dolphins (*Tursiops truncatus*) from the Canary Islands Suggests a Key Population for Conservation with High Connectivity within the North-East Atlantic Ocean

**DOI:** 10.3390/ani14060901

**Published:** 2024-03-14

**Authors:** Daniel A. Gómez-Lobo, Agustín P. Monteoliva, Antonio Fernandez, Manuel Arbelo, Jesús de la Fuente, Mónica Pérez-Gil, Nuria Varo-Cruz, Antonella Servidio, Enrique Pérez-Gil, Yaisel J. Borrell, Laura Miralles

**Affiliations:** 1Department of Functional Biology, University of Oviedo, 33006 Oviedo, Spain; dagomezlobo@uc.cl (D.A.G.-L.); borrellyaisel@uniovi.es (Y.J.B.); 2Department of Environmental Genetics, Ecohydros, 39600 Maliaño, Spain; apmonteoliva@ecohydros.com; 3Veterinary Histology and Pathology, Atlantic Center for Cetacean Research, Institute of Animal Health and Food Safety (IUSA), Veterinary School, University of Las Palmas de Gran Canaria, 35001 Las Palmas, Spain; antonio.fernandez@ulpgc.es (A.F.); manuel.arbelo@ulpgc.es (M.A.); jesus.delafuente@ulpgc.es (J.d.l.F.); 4Cetaceans and Marine Research Institute of the Canary Islands (CEAMAR), 35509 Las Palmas, Spain; monica@ceamar.org (M.P.-G.); nuriavaro@hotmail.com (N.V.-C.); antonella@ceamar.org (A.S.); kikelanza@gmail.com (E.P.-G.)

**Keywords:** *Tursiops truncatus*, Canary Islands, mitochondrial DNA, D-loop, special areas of conservation, conservation genetics, ecotypes

## Abstract

**Simple Summary:**

The common bottlenose dolphin, *Tursiops truncatus*, is a worldwide cetacean species essential for marine ecosystems’ health and balance. Understanding the genetic connectivity and structure of different populations is crucial for the correct management and conservation of a species, such as designing Special Areas of Conservation or Marine Protected Areas. In this study, we described the genetic composition of 49 bottlenose dolphins from the Canary Islands, which were previously unstudied, and compared them with individuals from the rest of the North-East Atlantic Ocean. The results showed that Canarian bottlenose dolphins have a remarkably diverse genetic composition, and this population is possibly part of a larger oceanic population in the North Atlantic. Therefore, the studied Special Areas of Conservation in the Canary Islands may correspond to a hotspot of genetic diversity and could be a strategic area for the conservation of the species.

**Abstract:**

In recent decades, worldwide cetacean species have been protected, but they are still threatened. The bottlenose dolphin (*Tursiops truncatus*) is a vulnerable keystone species and a useful bioindicator of the health and balance of marine ecosystems in oceans all over the world. The genetic structure of the species is shaped by their niche specialization (along with other factors), leading to the classification of two ecotypes: coastal and pelagic. In this study, the genetic diversity, population structure, and ecotypes of bottlenose dolphins from the Canary Islands were assessed through the analysis of 49 new samples from biopsies and from stranded animals using the 636 bp portion of the mitochondrial control region and 343 individuals from databases (*n* = 392). The results reveal high genetic diversity in Canarian bottlenose dolphins (Hd = 0.969 and π = 0.0165) and the apparent lack of population genetic structure within this archipelago. High genetic structure (*F*st, *Φ*st) was found between the Canary Islands and coastal populations, while little to no structure was found with the pelagic populations. These results suggest that Canarian bottlenose dolphins are part of pelagic ecotype populations in the North Atlantic. The studied Special Areas of Conservation in the Canary Islands may correspond to a hotspot of genetic diversity of the species and could be a strategic area for the conservation of the oceanic ecotype of bottlenose dolphins.

## 1. Introduction

The common bottlenose dolphin (*Tursiops truncatus*) is one of the most widely distributed cetacean species occurring in temperate and tropical waters worldwide [1]. As top predators, they are useful bioindicators of the health and status of marine ecosystems and play vital roles in maintaining the balance in such environments [2,3]. Thanks to their behavioral and ecological plasticity, bottlenose dolphins can inhabit a vast range of aquatic ecosystems, from deep oceanic waters to coastal estuarine ecosystems, even roaming into rivers [4,5]. This great ecological variability and the lack of apparent physical barriers to dispersal or gene flow in the marine environment make it challenging to define a population (stock) and its boundaries, which have important implications in both evolutionary and conservation biology.

The bottlenose dolphin is protected by the EU Habitats Directive 92/43/EEC. It is included in the Berne Convention as strictly protected fauna, and its coastal ecotype is present in the ACCOBAMS (Agreement on the Conservation of Cetaceans in the Black Sea, Mediterranean Sea and contiguous Atlantic area). In Spain, it is cataloged as Vulnerable (VU) in the National Catalog of Endangered Species in both peninsular waters (Order of 10 March 2000) and in those of the Canary Islands (Order of 9 June 1999). The bottlenose dolphin is also included as Vulnerable in the Red Book of Vertebrates, both in the waters of the European Union and in the Spanish Mediterranean. However, globally, the common bottlenose dolphin is cataloged as Least Concern (LC) in the IUCN Red List of Threatened Species [6], and different populations from distant geographical areas face different anthropogenic threats; therefore, such populations should be categorized and managed separately. For example, Mediterranean populations were classified as Vulnerable (VU) until 2021 [7], and currently, the Fiordland subpopulation in New Zealand is listed as Critically Endangered (CE) [8], raising special conservation concerns for small and resident coastal populations.

Coastal bottlenose dolphin populations are commonly found in shallow waters less than 40 m deep, while pelagic populations are observed in outer deeper oceanic waters (200 to 4000 m) [9,10], and several studies have noted differences in their distribution, diet, and skull morphology [11,12,13,14,15], leading to the idea of two different ecotypes. In addition, findings of significant genetic structure have reinforced this idea, with coastal populations presenting lower genetic diversity [16,17,18,19,20,21,22,23]. Site fidelity, along with resource specialization and different social and behavioral strategies, appears to be a strong driving force of genetic structuring in coastal resident bottlenose dolphins worldwide [12,15,16,19,20,21,23,24,25,26]. In the North Atlantic Ocean, pelagic populations show a highly diverse pattern with high levels of gene flow among extremely distant geographical regions, suggesting the existence of a single large panmictic oceanic population [16,18,20,22]. On the contrary, some coastal populations present fine-scale levels of genetic structure with low diversity [19,22,23,25,27], and even the recent extinction of a genetically isolated population (e.g., Humber estuary, UK) has occurred with no signs of repopulation so far [28]. Such contrasting patterns and the reduced population size of coastal bottlenose dolphins raise special concerns about the conservation of the species. Since coastal cetaceans could face more anthropogenic threats than oceanic ones [29], and their low effective population sizes might lead to a decrease in the adaptive potential to environmental changes [30,31], the identification of such threatened populations is crucial for the management of the species (e.g., designation of Special Areas of Conservation, SACs).

The Canary Islands is one of the major four archipelagos (Azores, Madeira, Canaries, and Cape Verde) within the Macaronesian region. This region is characterized by complex geomorphology, with several sea mountains, volcanic activity, and a rugged coastline [32]. Its bathymetry is typical of oceanic islands, rapidly reaching depths of 200 m near the coast (Figure 1). Many cetacean species inhabit and roam these oceanic waters, representing not only a hotspot of cetacean abundance and diversity [33] but also an important biological corridor for these large marine mammals due to their high dispersal capacities. Bottlenose dolphin populations observed in different SACs from the Canary Islands show high site fidelity patterns and are greater than populations of other archipelagos (e.g., Hawaii or Bahamas; [34]). Several individuals have been resighted off two or more islands and even in other archipelagos (Madeira), providing evidence of the long-distance movements (≈500 km) that these dolphins can undertake [34,35]. Nevertheless, to date, these populations remain unstudied in terms of genetic structure, connectivity with other regions, and ecotype assessment.

In this context, the aims of this study were to determine the population structure within the Canary Islands and to assess the ecotype of bottlenose dolphins using molecular markers. Moreover, data relating to North-East Atlantic Ocean (NEAO) bottlenose dolphins were added to our analysis to study the phylogeographic relationships and possible connectivity with the other populations from the North-East Atlantic basin. Since no levels of genetic structure were found in other archipelagos (within and between Azores and Madeira) of the Macaronesian region [18], the high dispersal of Canarian bottlenose dolphins [34,35], and the broad connectivity of the pelagic ecotype in the North-East Atlantic [18,20], we hypothesized that none or negligible levels of genetic structure should be observed within the Canary Islands, clustering with the pelagic ecotype of the North-East Atlantic.

## 2. Materials and Methods

### 2.1. Sample Collection

A total of 49 bottlenose dolphin samples were collected from the Canary Islands from 2005 to 2022 (Appendix A). Thirty-one biopsy samples were obtained from wild specimens in two different SACs (see studied area in Figure 1), the marine strips of Santiago-Valle Gran Rey in La Gomera island (ZEC-ES7020123) and Teno-Rasca in Tenerife Island (ZEC-ES7020017). The tissue size was 8 mm in diameter and length, and only adults were sampled. Photo identification was carried out simultaneously to spot individuals with site fidelity. Eighteen samples were obtained from stranded animals from five different locations (La Gomera, Tenerife, Gran Canaria, Fuerteventura, and Lanzarote). All stranding events were of single individuals (see Appendix A), and the individuals were in the fresh decomposition stage (recently dead individuals), ensuring that death occurred near the coast. Tissue samples were either immediately preserved in ethanol or first frozen at −20 °C and later placed in ethanol for long-term storage.

### 2.2. Genetic Analysis: Population Structure and Diversity in the Canary Islands

DNA was extracted from the skin samples using DNeasy Blood and Tissue Kit (QIAGEN, Venlo, The Netherlands), following the manufacturer’s instructions with modifications for small size samples (biopsies), such as longer lysis incubation (24 h), longer pre-elution incubation (5–10 min) and smaller elution volume (75 μL). All individuals were genetically sexed with the multiplexed SRY gene and ZFY/ZFX gene PCR [36]. A fragment of the mtDNA D-loop region was amplified using the primers described in the work of Dalebout et al. [37] following the protocol of Miralles et al. [38]. PCR sequencing in forward and reverse directions was carried out at Macrogen Inc. (Madrid, Spain) with a 3730XL DNA Analyzer (Applied Biosystems, Waltham, MA, USA). All of the obtained sequences were visualized, assembled, and checked for ambiguities in BioEdit, Version 7.0.5.3 [39]. The sequences were aligned and manually edited in BioEdit, producing a dataset of 49 sequences. Prior to molecular analyses, all of the sequences were corroborated using the Nucleotide BLAST tool (Basic Local Alignment Search tool, NCBI).

Genetic diversity and structure (*F*st, *Φ*st) indexes were assessed using ARLEQUIN, Version 3.5.1.2 [40]. *F*st may be an indicator of short-term or recent population processes, while *Φ*st may be an indicator of longer-term or older processes. Therefore, it is useful to calculate both types of indexes for any data set. Combining these statistics will enable more robust analyses of population structure than what is possible with only *F*st. Moreover, if they are different, it is possible that sample size and mutations have a larger influence on the results obtained. ARLEQUIN was also used to estimate the number of segregating sites (*S*), haplotypes (Nh), unique haplotypes (h), haplotype diversity (Hd), nucleotide diversity (π), and the average number of nucleotide differences between pairs of sequences (*k*). To determine if there were any deviations from the Wright–Fisher mutation-drift equilibrium due to population bottlenecks or expansions, Fu’s Fs and Tajima’s D indices were calculated in ARLEQUIN with their respective *p* values.

Phylogenetic relationships among the different haplotypes were inferred from a median-joining network constructed with PopArt, Version 1.7 [41,42], with the homoplasy parameter (ε) set to zero. To further visualize the possible genetic structure within the Canary Islands, non-metric Multidimensional Scaling (nMDS) analysis was conducted in PAST, Version 4.03 [43], using the mutation distribution of haplotypes and applying Tamura [44] for genetic distances and considering tolerable stress values <0.2 [45].

### 2.3. Genetic Analysis: Population Structure and Diversity in the NEAO

To study the phylogeographic relationships of the Canary Islands within the NEAO, the complete dataset of Louise et al. [20], except for the individuals of unknown origin, was downloaded from GenBank (*n* = 343). This dataset comprised four main groups containing several regions from the NEAO and Mediterranean Sea: the coastal south group (English Channel, Arcachon estuary, and South Galicia bottlenose dolphins), coastal north group (UK and Ireland resident or mobile coastal groups), pelagic Atlantic group (Azores archipelago and Bay of Biscay), and finally, the pelagic Mediterranean group (Gulf of Cadiz and Corsica) (see Louise et al. [20] for detailed description). The sequences were aligned using the Clustal W tool within *MEGA-X*, Version 10.0.5 [46], and trimmed to 636 pb to match our dataset. Since no polymorphism was present within the trimmed regions, no haplotype or information was lost, producing a final dataset of 392 sequences and defining 70 haplotypes, including the Canary Islands from this study. Genetic diversity and structure (*F*st, *Φ*st) indexes were calculated in ARLEQUIN in addition to an analysis of molecular variance (AMOVA) with 10,000 permutations. For *Φ*st, the best-fit model of molecular evolution was determined using *MEGA-X*, which resulted in T92 +G + I [44], based on the Bayesian Information Criterion (BIC; [47]), with a gamma value of 0.46. Finally, a haplotype network was constructed using the median-joining algorithm in PopArt with ε set to zero.

## 3. Results

### 3.1. Population Structure and Genetic Diversity in the Canary Islands

In total, 49 mtDNA sequences of 636 bp were obtained, defining 28 haplotypes across the Canary Islands; 29 individuals corresponded to previously reported haplotypes, and 20 individuals presented 15 new unreported haplotypes (CAN1-CAN15, Appendix A). New haplotypes were uploaded to GenBank under the accession numbers OQ656769-OQ656783.

Overall, mitochondrial haplotype and nucleotide diversities were high: Hd = 0.969 and π = 0.0165 (Table 1). Tenerife presented the highest number of haplotypes (Nh), unique haplotypes (h), and segregating sites (*S*), but it was also the location with the highest sample size (*n* = 27). Similar values of genetic variability in both largest samples in terms of π, Hd, and K were obtained despite the smaller sample size of La Gomera in comparison with Tenerife (Hd = 0.955 and 0.952, respectively) (Table 1). However, no significant population structure was found between these two localities (*F*st = 0.0008, *Φ*st = 0.014; *p* > 0.2). In addition, no differences were found between biopsies and stranding samples (*F*st = 0.005, *Φ*st = 0.049 *p* > 0.05), and all diversity indexes presented high similarity (Appendix A), discarding possible confounding effects between the two types of samples. Both Fu’s Fs (Fs = −5.88, *p* = 0.052) and Tajima’s D (D = 0.41, *p* = 0.725) were not significant.

The median-joining network (Figure 2A) shows a highly diverse and reticulated pattern, with most individuals forming single haplotypes with multiple mutational steps. The two more distant haplotypes were separated by 47 bp. Only eight haplotypes were shared between individuals from different locations, where six were shared between Tenerife and La Gomera, one between Fuerteventura and La Gomera, and one between Gran Canaria and Lanzarote.

Non-metric Multidimensional Scaling analysis shows the lack of genetic structure within the Canary Islands since no clear clustering is observed, and all samples are scattered across the plot. The low stress value (0.09) indicates the validity of the analysis (Figure 3).

### 3.2. Population Structure and Genetic Diversity in the Canary Islands

A dataset of 392 mtDNA sequences was obtained by combining this study (*n* = 49) and the work of Louis et al. [20] (*n* = 343), defining 70 haplotypes. With the inclusion of the highly diverse Canary population, the overall haplotypic diversity was augmented (Hd = 0.905) in relation to the values previously reported by Louis et al. [20] (Hd = 0.883). Out of the 70 haplotypes, 17 were private for the Canary Islands, being the second population with the most unique haplotypes after the Pelagic Atlantic samples (h = 25, Table 2). Despite having the smallest sample size, the Canary Islands presented the highest haplotypic diversity (Hd = 0.969) (Table 1 and Table 2). Initially, an analysis of molecular variance (AMOVA) was tested with the Canary Islands as an independent group against pelagic and coastal populations, resulted in being not significant in the global structure (*Φ*ct = 0.377, *p* = 0.139; *Fc*t = 0.075, *p* = 0.268), which is likely due to the large variability within the populations (*Φ*st = 0.434, *p* < 0.0001; *F*st = 0.219, *p* < 0.0001). A second test was performed, grouping the Canary Islands within the pelagic group, which also resulted in no significance (*Φ*ct = 0.436, *p* = 0.103; *Fc*t = 0.136, *p* = 0.103). The pairwise comparisons of *F*st and *Φ*st obtained by Louis et al. [20] were replicated with the addition of the Canary Islands population, where the last was mainly differentiated from the coastal populations (Table 3). All of the comparisons among the Canary Islands and coastal populations were significant, with high *Fs*t and *Φ*st values (*p* < 0.001), while no structure was found when compared with the pelagic Atlantic populations (*F*st and *Φ*st values). However, one significant but low level of genetic structure (*Fs*t = 0.057, *p* < 0.001) was found between the Canary Islands and the pelagic Mediterranean (but not the *Φ*st value). It is the only comparison when *F*st is significant but *Φ*st is not.

A global haplotype network including all sequences from Louis et al. [20] was performed with the addition of the Canary Islands sequences (Figure 2B). All of the individuals clustered among the pelagic haplotypes in the upper part of the network, except for one stranded individual (CET0564), which showed the haplotype Ttrunc2, typical of the coastal bottlenose dolphins. It is worth mentioning that, in the upper-left side of the network, a coastal north haplotype (in red) from the UK and Ireland clustered with several pelagic haplotypes from the Atlantic (Azores and Bay of Biscay) and Mediterranean (Gulf of Cadiz and Corsica) as well as five Canarian haplotypes in a branch with a star-like pattern. In general, branches of the network were well defined in terms of several mutations between the closest haplotypes (e.g., four, five or seven positions).

## 4. Discussion

The genetic identification of natural populations is of crucial importance for the correct management and risk assessment of a species since small isolated populations are at increased risk of the effects of genetic drift and inbreeding [30], which can increase extinction probability. This is especially true in the case of bottlenose dolphins because coastal populations have been described to have low levels of genetic diversity, and even the extinction of an isolated population has been reported (e.g., Humber Estuary, UK) [28]. The results of this study would help to define key areas within the Macaronesian region for the management and long-term conservation of this relevant marine species protected in Europe under the EU Habitats Directive (92/43/CEE).

This study is the first to report the genetic structure of the population within the Canary Islands and to assess the ecotypes using molecular markers (i.e., mtDNA) and biopsies of free-ranging individuals. The overall mitochondrial haplotypic diversity found in this study (Hd = 0.969) is the highest reported in any previously studied bottlenose dolphin population in the North Atlantic [17,18,20,27]. Bottlenose dolphins from the Canary Islands were found to be remarkably diverse, with high genetic diversity indexes (Table 1). From a total of 49 samples, we found 28 haplotypes, meaning that more than half of the individuals sampled presented a different mtDNA sequence with multiple mutations between them (overall *k* > 10). Both Fu’s Fs and Tajima’s D were not significant, suggesting a relatively stable population size under mutation–drift equilibrium. No sign of genetic structuring among the islands was found in this work (Figure 2A). The haplotype network shows both patterns of high genetic diversity and the lack of a fine-scale structure, showing three major characteristics: (1) the presence of many haplotypes composed of single individuals, (2) multiple mutational steps among haplotypes, and (3) samples from different localities scattered across the network (Figure 2A). Additionally, non-metric Multidimensional Scaling analysis reinforced the evidence of a lack of genetic structure within the Canary Islands since no clear clustering is observed and all samples are scattered across the plot (stress value = 0.09; Figure 3). Although the lack of structure was expected, the small sample size of Lanzarote, Gran Canaria, and Fuerteventura, coupled with the absence of nuclear markers (microsatellites), might hinder the signals of a fine-scale genetic structure.

Previously, only Fernández et al. [27] reported genetic data of six stranded bottlenose dolphins from the Canary Islands. These authors found high nuclear and mitochondrial diversity. In our study, samples from the Canary Islands were grouped with the Azores, Basque Country, and Mainland Portugal, forming an offshore population in contrast with the genetically isolated population of Southern Galicia and the Sado estuary [27]. As reported in the Azores and Madeira archipelagos [18], along with photo identification data [34,35], our results support the hypothesis of the absence of a fine-scale genetic structure within the Canary Islands, with this population possibly grouping within the diverse large oceanic ecotypes.

The global haplotype network indicated that individuals from the Canary Islands are closely related to both pelagic Atlantic and pelagic Mediterranean populations by clustering within the upper pelagic mitochondrial lineage (Figure 2B). All of the individuals clustered among the pelagic haplotypes in the upper part of the network, except for one stranded individual (CET0564), showing haplotype Ttrunc2, which is typical of coastal populations. The Canary samples were scattered across the network, sharing ten and three haplotypes with pelagic Atlantic and pelagic Mediterranean populations, respectively, which could indicate current or historical gene flow, incomplete lineage sorting, or introgression [20]. In addition, despite having less than half of the sample size of the pelagic Atlantic population, the population of the Canary Islands possessed a remarkably high number of seventeen private haplotypes (i.e., haplotypes only found in that locality) in comparison to twenty-five (Table 2). The lack of genetic structure with pelagic populations, the deep bathymetry of the islands, and the high levels of haplotypic diversity support the hypothesis that bottlenose dolphins from the Canary Islands are part of a large oceanic population in the North-East Atlantic [18,20]. This connectivity among populations could be maintained by the high dispersal capacity of the species [35,48,49] and adaptations to deep oceanic environments [21]. However, one low but significant value in terms of genetic structure (*F*st = 0.057, *p* < 0.001) was found between the Canary Islands and pelagic Mediterranean (but not the *Φ*st value) (Table 3). It is known that the *F*st method is largely influenced by the presence of rare variants [50], while *Φ*st statistics are not. *Φ*st is derived from two different statistical distributions: the distribution of allele (haplotypes) frequencies among populations and the distribution of evolutionary distances among alleles [51]. When the significance of both markers differed, it is possible that samples size and/or mutation had a larger influence on the results obtained. After a population splits and until subpopulations have reached a stable equilibrium, *F*st is likely to increase first, indicating recent events. Only after new alleles have arisen and monophyletic clades of alleles have begun to arise in different subpopulations will *Φ*st begin to increase substantially [51]. This way, it takes advantage of this additional information and provides greater insight into the patterns of relationships among the populations.

The results obtained in this work are in concordance with those obtained by Hildebrandt (unpub. data; [52]), in which Canarian bottlenose dolphins showed high diversity indexes, a lack of structure, and similarities with bottlenose dolphins from the North Atlantic Ocean. The Canary Islands are considered a hotspot of cetacean biodiversity [33], one of the most diverse places for cetaceans and the largest in Europe [53]. However, just three species dominated the sightings: bottlenose dolphins, pilot whales (*Globicephala macrorhynchus*) and spotted dolphins (*Stenella frontalis*) [53]. Comparing the results obtained here with these other two delphinid species, we observed the same lack of genetic structuring across the Canary Islands in spotted dolphins [54] but not in pilot whales [52]. On a broader scale, it has been described that spotted dolphins represent several distinct units in the Atlantic Ocean: Macaronesian group clustering, Canary Islands, Azores and Madeiran individuals [54].

Bottlenose dolphins are a highly endangered species due to coastal activities and fisheries. They are protected in Europe under the EU Habitats Directive (92/43/CEE), the Berne Convention and the ACCOBAMS, which requests the designation of SACs for their protection. Our results highlight the importance of the SACs in terms of managing and preserving bottlenose dolphins inhabiting the Canary Islands since this region seems to represent a hotspot of genetic diversity for a large pelagic population. The protection of these strategic areas could have positive impacts even in the outer parts of the marine reserve [23,55] thanks to the high connectivity of such pelagic ecotypes in the North-East Atlantic Ocean. This study provides baseline data for further investigations of the fine-scale genetic structure within the Canarian and Macaronesian region. Future studies that include nuclear markers (microsatellites) or genomics would provide higher-resolution information [56] on the connectivity among islands and detailed information for the future management of this protected species.

## 5. Conclusions

The analysis of 49 new samples, along with 343 individuals from databases, revealed a remarkable level of genetic diversity among Canarian bottlenose dolphins, as indicated by the highest reported mitochondrial haplotypic diversity in any North-East Atlantic bottlenose dolphin population. In line with our hypothesis, we found negligible levels of genetic structure within the Canary Islands, suggesting a cohesive population across the archipelago. The results align with the absence of fine-scale genetic structure reported in other oceanic archipelagos and support the hypothesis that Canarian bottlenose dolphins are part of a larger oceanic population in the North-East Atlantic.

Results from this research highlight the importance of Special Areas of Conservation (SACs) in the Canary Islands. The designation of SACs is crucial for preserving the genetic diversity of bottlenose dolphins, particularly considering their classification as a strategic area for the conservation of the oceanic ecotype. Additionally, we highlight the importance of incorporating nuclear markers (microsatellites) or SNPs to enhance the resolution of connectivity and provide detailed information for the ongoing conservation and management of this protected species.

## Figures and Tables

**Figure 1 animals-14-00901-f001:**
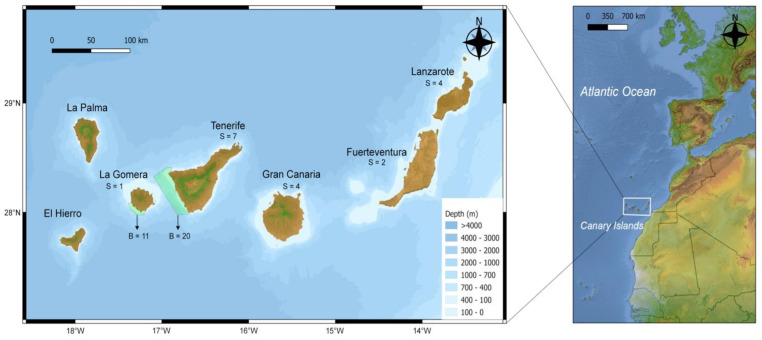
Map of the Canary Islands with sampling scheme of stranded (S) individuals and biopsy (B) samples. Areas in green highlight the Special Areas of Conservation (SACs) in La Gomera (ZEC-ES7020123) and Tenerife (ZEC-ES7020017) where biopsies were collected. Isobaths are plotted and denoted on a scale of blue.

**Figure 2 animals-14-00901-f002:**
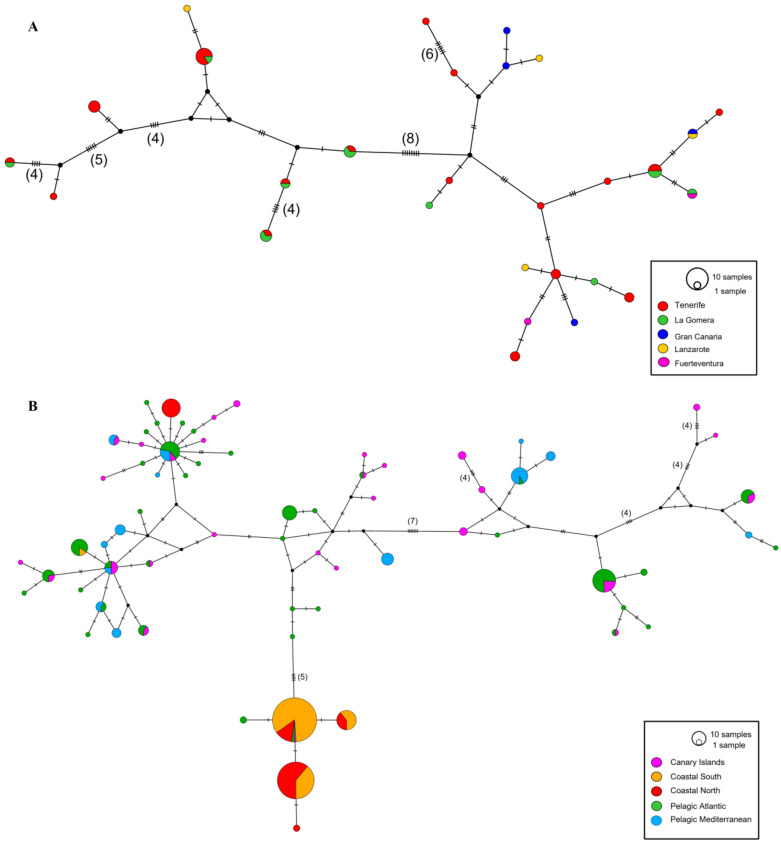
Median-joining network based on 636 bp of mtDNA haplotypes found in bottlenose dolphins from the Canary Islands (**A**), and from the North-East Atlantic and Mediterranean (**B**). Each circle represents a unique haplotype colored proportionally to the amount of individuals found in each location. Size of the circle is proportional to the haplotype frequencies. Black circles represent unsampled or extinct intermediate haplotypes. Hatch marks represent mutational steps. More than 3 mutational steps are denoted in parenthesis.

**Figure 3 animals-14-00901-f003:**
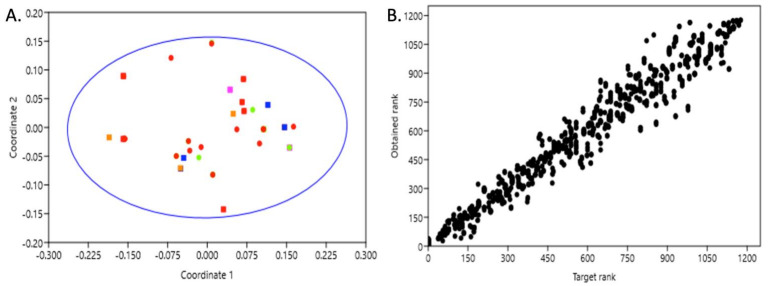
Non-metric Multidimensional Scaling of the five localities sampled in the Canary Islands. (**A**) Scatter plot showing necropsies (squares) and biopsies (circles) samples colored depending on the locality of collection. Red = Tenerife; Green = La Gomera; Blue = Gran Canaria; Orange = Lanzarote; Pink = Fuerteventura. Here, 95% ellipse is denoted with in a blue line. (**B**) Shepard plot. The stress value of the Shepard plot is 0.09.

**Table 1 animals-14-00901-t001:** *Bottlenose dolphin* mitochondrial genetic diversity found in bottlenose dolphins from the Canary Islands, including sample size (*n*), segregating sites (*S*), number of haplotypes (Nh), number of unique haplotypes (h), haplotype diversity (Hd), nucleotide diversity (π), and average number of nucleotide differences (*k*). SD = standard deviation.

Populations	*n*	*S*	Nh	h	Hd (SD)	π (SD)	*k*
Tenerife	27	35	17	11	0.952 (0.025)	0.0166 (0.009)	10.571
La Gomera	12	31	9	2	0.955 (0.047)	0.0153 (0.008)	9.713
Lanzarote	4	24	4	3	1.000 (0.177)	0.0206 (0.014)	13.121
Gran Canaria	4	16	4	3	1.000 (0.177)	0.0149 (0.010)	9.491
Fuerteventura	2	9	2	1	1.000 (0.500)	0.0128 (0.014)	8.122
Total	49	43	28	/	0.969 (0.011)	0.0165 (0.009)	10.509

**Table 2 animals-14-00901-t002:** *Bottlenose dolphin* mitochondrial genetic diversity in the North-East Atlantic, including sample size (*n*), segregating sites (*S*), number of haplotypes (Nh), number of unique haplotypes (h), haplotype diversity (Hd), nucleotide diversity (π), and average number of nucleotide differences (*k*). Data from this work and from Louis et al. [20]. SD = standard deviation.

Populations	*n*	*S*	Nh	h	Hd (SD)	π (SD)	*k*
Canary Islands	49	43	28	17	0.969 (0.011)	0.0165 (0.009)	10.503
Coastal south	115	12	4	0	0.499 (0.044)	0.0014 (0.001)	0.889
Coastal north	76	13	5	2	0.667 (0.042)	0.0063 (0.003)	4.028
Pelagic Atlantic	101	41	38	25	0.929 (0.013)	0.0155 (0.007)	9.881
Pelagic Mediterranean	51	28	15	8	0.902 (0.022)	0.0137 (0.007)	8.680
Total	392	56	70	/	0.905 (0.009)	0.0140 (0.007)	8.894

**Table 3 animals-14-00901-t003:** Population pairwise *F*st (above diagonal) and *Φ*st (below diagonal) values in terms of bottlenose dolphins from the Canary Islands.

Populations	Canary Islands	Coastal South	Coastal North	Pelagic Atlantic	Pelagic Mediterranean
Canary Islands	-	0.291 **	0.191 **	0.015	0.057 **
Coastal south	0.635 **	-	0.252 **	0.279 **	0.328 **
Coastal north	0.401 **	0.233 **	-	0.195 **	0.222 **
Pelagic Atlantic	0.004	0.541 **	0.349 **	-	0.071 **
Pelagic Mediterranean	0.040	0.671 **	0.446 **	0.056 **	-

** *p* < 0.01 after sequential Bonferroni correction.

## Data Availability

Data are contained within the article and Appendix A. New haplotypes were accessible in GenBank under the accession numbers OQ656769-OQ656783.

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
