# Peer review of "Mitochondrial Variation of Bottlenose Dolphins (Tursiops truncatus) from the Canary Islands Suggests a Key Population for Conservation with High Connectivity within the North-East Atlantic Ocean"

_animals, 2024, doi:10.3390/ani14060901_

Round 1
Reviewer 1 Report
Comments and Suggestions for Authors
This is a very good manuscript presenting information on mtDNA diversity for Canary Island bottlenose dolphins. I think the study is sound and well presented. I think the analyses would benefit of further comparisons with sequences from other areas in the Atlantic, for example the Caribbean Sea and the Western North Atlantic. Bottlenose dolphins can move long distances and their genetic structure is complex, therefore a more thorough analyses would improve the quality of the paper. Also, it is important that the authors comment more thorhougly on the Phist and Fst results.

English can be improved a little bit by a native reader
Author Response
R1: Comments and Suggestions for Authors:
This is a very good manuscript presenting information on mtDNA diversity for Canary Island bottlenose dolphins. I think the study is sound and well presented. I think the analyses would benefit of further comparisons with sequences from other areas in the Atlantic, for example the Caribbean Sea and the Western North Atlantic. Bottlenose dolphins can move long distances and their genetic structure is complex, therefore a more thorough analyses would improve the quality of the paper. Also, it is important that the authors comment more thorhougly on the Phist and Fst results.
AUTHORS: Thank you very much for your comments and suggestions. We made a more detailed and thorough discussion about Phist and Fst (adding information about it in Material and Methods, Results and Discussion sections) and we made all the changes that you suggested except one. It is not possible to add sequences from Western North Atlantic and Caribbean Sea without loosing crucial information. We tried this before submitting, at the time when we were making our initial analyses but two main facts stopped us. Haplotypes frequencies were lacking, and worst, after alignments the sequences dataset needed to be drastically cut provoking the loss of many new Canarian haplotypes, if we do so. Our sequences, and also the sequences’ dataset from Louis et al (2014), were successfully analyzed and trimmed to 636 bp long, but when including the Western North Atlantic and Caribbean Sea we need to trim the dataset to only 324 bp (it mean to loose aprox. 50% data). So, all this together including the lack of haplotype frequencies from Western North Atlantic and Caribbean Sea would make a poor network with drastic lack of information. Dolphin research is a hot topic and we really hope that soon new sequences from that geographic area will be available and allows to have a more complete picture about the global species genetic diversity distribution. We would like to be able to do it now, but unfortunately it’s not possible without loosing the novelty of this research.
Specific changes from PDF:
Fig.1: show in the map the location of the Canary islandas compared to the rest of Europe or the North east Atlantic
AUTHORS: Done.
Line 152: you did not use a software to check the raw sequences? Geneious for example?
AUTHORS: Yes, we used BioEdit. It is described in the text as follow: “All obtained sequences were visualized, assembled, and checked for ambiguities in BioEdit, Version 7.0.5.3 [38]. Sequences were aligned and manually edited in BioEdit,”
Line 184: it is great that you included so many information (sequences or haplotuypes in your analyses) however, for completenes and considering bottlenose dolphins are able to move long distances, I think it would not hurt to include some sequences from the Northwest Ataltinc and the Caribbean and even from Brazil. i know it is a pain to redo these analyses but it may be worth it.
AUTHORS: Thank you for this comment. Indeed, it was worthy to explore it (and we did it). However, it was impossible to do it without loosing essential information.
We amplified a 636 bp sequence describing 15 new haplotypes from the Canary Islands (from the 28 haplotypes we found). We compared our results with the sequences from Louis et al. (2014) - a very good complete dataset of 343 sequences. Our sequences aligned perfectly with their sequences and we do not need to cut or loose any information. We kept the 636 bp dataset. However, when we added sequences from Western North Atlantic and Caribbean Sea we need to trim all of them to 324 bp (it mean to loose aprox. 50% data). Apart from loosing number of base pairs, many haplotypes from the Atlantic, Mediterranean and the new one from Canary Islands disappeared too. So, the new sequences from our study needed to be drastically cut loosing a lot of information.
Furthermore, the comparison between our dataset and the one from Louis et al (2014) included the information about haplotype frequencies and allow visualizing it in the network, for all the dataset. However, it was not possible to retrieve the haplotype frequencies from all Western North Atlantic and Caribbean Sea datasets. In other words, adding Western North Atlantic and Caribbean Sea impoverish analyses in a significant way. Louis et al (2014) constructed the comparative network without frequencies and they also reported the lost of haplotypes after trimming. Then, taking into account the lost of crucial information (many of new described haplotypes from Canary Islands due to trimming and the absence of Western North Atlantic and Caribbean Sea haplotype frequencies) we decided to excluded them for the analyses. Dolphins studies are a hot topic and soon (we hope that and it’s in our focus of research priorities and attention as well) this will be possible.
Fig.2: include haplos from the Caribbean and NWA
AUTHORS: It is not done and explained in the previous point.
Line 246: resulted as non significant
AUTHORS: Done.
Line 255: what about results from the Phist? analyse how differents are the haplotypes in terms of their nucleotide differences and comment on that
AUTHORS: Modified in the manuscript. Results of Φst included here and following your recommendation, those results were analysed in detail in discussion. Also, more detailed information about this two indexes was included in material and methods.
Text included in discussion:
“However, It was found a significant Fst value for genetic structuring between the Canary Islands and Pelagic Mediterranean (Fst = 0.057, p < 0.001) (but not for the Φst value) suggesting a probable recent structuring patterns (Table 3). It is known that the Fst method is largely influenced by the presence of rare variants [50] while Φst statistics does not. The Φst is derived from two different statistical distributions: the distribution of allele (haplotypes) frequencies among populations and the distribution of evolutionary distances among alleles [51]. When significance of both markers differed, it is possible that samples sizes and/or mutation have a larger influence on the results obtained. After population splits and until subpopulations have reached a stable equilibrium, Fst is likely to increase first, indicating recent events. Only after new alleles have arisen, and monophyletic clades of alleles have begun to arise in different subpopulations, Φst will begin to increase substantially [51].”
Text included in Material and Methods:
“Genetic diversity and structure (Fst, Φst) indexes was assessed in ARLEQUIN, Version 3.5.1.2 [40]. Fst may be an indicator of short-term or recent population processes, while Φst may be an indicator of longer-term or older processes. So, it is useful to calculate both types of indexes for any data set. Combining these statistics will enable more robust analyses of population structure than what is possible with only Fst. Moreover, if they are different its possible that samples sizes and mutation have a larger influence on the results obtained.”
Line 332: do not ,limit your discussion to only bottlnose dolphins. Compare this results to for example other delphinid species that are founf in the region and for whom genetic data mtDNA is availanble, for example pilot whales and stenella frontalis
AUTHORS: Thank you for this suggestion. We include a paragraph about the comparison between this three species in the revised version of the manuscript.
“Canary Islands are considered a hotspot of cetacean biodiversity [33], one of the most diverse places for cetaceans and the largest in Europe [53]. However, just three species dominated the sightings: bottlenose dolphins, pilot whales (Globicephala macrorhynchus) and spotted dophins (Stenella frontalis) [53]. Comparing the results obtained here with these other two delphinid species, we observed the same lack of genetic structuring across Canary Islands in spotted dolphins [54] but not in pilot whales [55]. In a broader scale, it has been described that spotted dolphins represented several distinct units in the Atlantic Ocean, being the Macaronesian group clustering Canary Islands, Azores and Madeira individuals [54]. “
Reviewer 2 Report
Comments and Suggestions for Authors
General remark 1: please do not use English and Latin names of bottlenose dolphin (Tursiops truncates) as synonyms in manuscript. At first mentioning use both names “bottlenose dolphin (Tursiops truncates)” and afterward use either English name or Latin name.
General remark 2: names of haplotypes are too complicated and too long “TtruncCAN1, Isolate BSEA6, Ttrunc7, voucher TT027”. I suggest to rename it that readers can trace the logic of haplotype naming, the naming can be related to “haplogroups”, the detection frequency, etc.
General remark 3: in headings Capitalize each word style should be used.
L24 change to “which were previously unstudied” avoiding term population, since the boundaries of certain population are not clear
L37, L155 636 bp
L100 (Figure 1) should be
L120-122 move to discussion or conclusions, this sentence does not belong to the introduction
L128, L224 please delete “Supplementary material”
L147-148 please specify what were modification for the DNA isolation, it can be essential for other researcher conducting similar investigations.
L148-149 the references here on the genetic sexing is missing
L149 please take into consideration that D-loop and control region are not synonyms
L156-160 please change haplotype names. In Table S1 change ordering of samples based on localities and different haplotypes; indicate which haplotypes are newly identified.
L186 please correct “636pb”
In methods do not cite Table 1, Table 3, Figure 2A, because you don't keep the order of in-text citation of figures/tables, also when you cite in-text figures/table they should be presented immediately. For instance Table 3 was mentioned before Table 2 and in page 5, while the Table 3 is presented in page 8.
L189 please indicate differences Fst and Φst, and how they were calculated. From the theoretical point of view Φst is better for haploidic mtDNA and Fst is better for diploidic DNA. What was the reason for calculating both parameters?
L198-209 how many haplotypes were newly identified, please include it in the text in this section
L201 “S” should not be in bold
Table 1. Please include also SE of π as it was done for Hd
L202-204 I do not support such interpretation of results, there is no difference between 0.952 ±0.025 and 0.955 ± 0.047. You should state that similar values of genetic variability in both largest samples in terms of π, Hd and K were obtained
Table S2 replace commas with dots
L205 p>0.05
L217 47 bp
I suggest to move Figure S1 to the main manuscript text, since it is important data. Also divide Figure to (a) and (b) parts, not left and right
L255-258 also the single red circle is in the upper part of the network, this should be mentioned.
CONCLUSIONS part is too long, please shorten it, some sentences such as L147-148 might me excluded.
Author Response
R2: Comments and Suggestions for Authors:
General remark 1: please do not use English and Latin names of bottlenose dolphin (Tursiops truncates) as synonyms in manuscript. At first mentioning use both names “bottlenose dolphin (Tursiops truncates)” and afterward use either English name or Latin name.
AUTHORS: Ok. We modified the text accordingly.
General remark 2: names of haplotypes are too complicated and too long “TtruncCAN1, Isolate BSEA6, Ttrunc7, voucher TT027”. I suggest to rename it that readers can trace the logic of haplotype naming, the naming can be related to “haplogroups”, the detection frequency, etc.
AUTHORS: Thank you for your comment. We shorted the name of the new haplotypes we described in this study to make them not so complicated and long (e.g. TtruncCAN1 as just CAN1). All of them (n=15) were named from CAN1 to CAN15 acording to the date of sampling. The oldest is CAN1 and the most recent is CAN15. Also, the names of haplotypes that were previously described and deposited on Genbank were shorted when possible (e.g. isolate BSEA6 as only BSEA6, or voucher TT027 as TT027) but the main name stays as their authors called previously and the same as referenced on Genbank.
General remark 3: in headings Capitalize each word style should be used.
AUTHORS: Ok. We modified the text accordingly.
Specific changes to be made:
L24 change to “which were previously unstudied” avoiding term population, since the boundaries of certain population are not clear
AUTHORS: OK, changed.
L37, L155 636 bp
AUTHORS: OK. Done.
L100 (Figure 1) should be
AUTHORS: OK. Changed.
L120-122 move to discussion or conclusions, this sentence does not belong to the introduction
AUTHORS: OK. It has been moved to discussion section.
L128, L224 please delete “Supplementary material”
AUTHORS: OK. Done.
L147-148 please specify what were modification for the DNA isolation, it can be essential for other researcher conducting similar investigations.
AUTHORS: OK. Done.
L148-149 the references here on the genetic sexing is missing
AUTHORS: Added.
L149 please take into consideration that D-loop and control region are not synonyms
AUTHORS: OK. Changed as “a fragment of the mtDNA D-loop region”
L156-160 please change haplotype names. In Table S1 change ordering of samples based on localities and different haplotypes; indicate which haplotypes are newly identified.
AUTHORS: Names were changes according to your previous comment. We follow your recommendation and we reorder them by location and putting the new haplotypes first. Also, following your recommendation, we put in bold the new haplotypes to easy identify them. However haplotype numeration cannot be modified for trazability between Genbank accession numbers and haplotypes’ codes therein and codes in the article. It has been explained in the text accordingly too.
L186 please correct “636pb”
AUTHORS: OK. Changed
In methods do not cite Table 1, Table 3, Figure 2A, because you don't keep the order of in-text citation of figures/tables, also when you cite in-text figures/table they should be presented immediately. For instance Table 3 was mentioned before Table 2 and in page 5, while the Table 3 is presented in page 8.
AUTHORS: OK. Changed
L189 please indicate differences Fst and Φst, and how they were calculated. From the theoretical point of view Φst is better for haploidic mtDNA and Fst is better for diploidic DNA. What was the reason for calculating both parameters?
AUTHORS: Genetic diversity and structure (Fst, Φst) indexes were calculated in ARLEQUIN [40] "Use conventional F-statistics" was used to estimate Fst. For Φst, we used "Compute a distance matrix" setting. The best-fit model of molecular evolution was determined using MEGA-X, which resulted in T92 +G +I [43], based on the Bayesian Information Criterion (BIC; [46]), with a gamma value of 0.46.
The most common use of F-statistics is to make inferences on demographic processes taking place within and among populations, such as migration, genetic drift, extinction and colonization. Therefore, the ideal summary statistic would provide information only on such demographic processes, and not about the purely genetic process of mutation (Ryman & Leimar 2008). The process of mutation is mostly a quality of the markers that are used, and in the great majority of cases, we are not interested in the markers per se, which are generally chosen to be selectively neutral (Meirmans and Hedrick, 2011). Sequence data are of a different nature than allelic data as they contain information on the evolutionary relationships between haplotypes. Simulations have shown that unlike Fst, Φst is independent of the mutation rate when calculated for sequence data (Kronholm et al. 2010) and at the same time Fst is not so reliable in connectivity analysis, especially at low sample size. However, this statistic has been used for several decades and, despite its shortcomings, continued use will allow a better comparison with those past studies, especially because reanalysis of the old data is mostly not possible (Neigel 2002). Combining these statistics will enable more robust analyses of population structure than what is possible with only Fst. If they are different its possible that samples sizes and mutation have a larger influence on the results obtained. It has been said that after a population splits, until subpopulations have reached a stable equilibrium (& for many organisms this is likely to not be the case at present). Then, Fst is likely to increase first, and only after new alleles have arisen, and monophyletic clades of alleles have begun to arise in different subpopulations, will Φst begin to increase substantially. Φst may be thought of as estimating the additional time since the common ancestry of randomly chosen alleles that accrues as a result of populations being subdivided, provided that the measure of evolutionary distance between any two alleles is proportional to the time since their most recent common ancestor (Holsinger and Weir, 2009). That is, Fst may be an indicator of short-term or recent population processes, while Φst may be an indicator of longer-term or older processes. So, it is useful to calculate both types for any data set.
We have added part of this explanation in both material and methods and in the discussion sections.
Text included in Material and Methods:
“Genetic diversity and structure (Fst, Φst) indexes was assessed in ARLEQUIN, Version 3.5.1.2 [40]. Fst may be an indicator of short-term or recent population processes, while Φst may be an indicator of longer-term or older processes. So, it is useful to calculate both types of indexes for any data set. Combining these statistics will enable more robust analyses of population structure than what is possible with only Fst. Moreover, if they are different its possible that samples sizes and mutation have a larger influence on the results obtained.”
Text included in discussion:
“However, It was found a significant Fst value for genetic structuring between the Canary Islands and Pelagic Mediterranean (Fst = 0.057, p < 0.001) (but not for the Φst value) suggesting a probable recent structuring patterns (Table 3). It is known that the Fst method is largely influenced by the presence of rare variants [50] while Φst statistics does not. The Φst is derived from two different statistical distributions: the distribution of allele (haplotypes) frequencies among populations and the distribution of evolutionary distances among alleles [51]. When significance of both markers differed, it is possible that samples sizes and/or mutation have a larger influence on the results obtained. After population splits and until subpopulations have reached a stable equilibrium, Fst is likely to increase first, indicating recent events. Only after new alleles have arisen, and monophyletic clades of alleles have begun to arise in different subpopulations, Φst will begin to increase substantially [51].”
L198-209 how many haplotypes were newly identified, please include it in the text in this section
AUTHORS: OK. Included.
L201 “S” should not be in bold
AUTHORS: OK. Changed
Table 1. Please include also SE of π as it was done for Hd
AUTHORS: OK. Done. Moreover, we have modified Table 2 accordingly, by adding it too.
L202-204 I do not support such interpretation of results, there is no difference between 0.952 ±0.025 and 0.955 ± 0.047. You should state that similar values of genetic variability in both largest samples in terms of π, Hd and K were obtained
AUTHORS: Thanks for your comment. We agree and consequently we modified in the text as you suggested.
Table S2 replace commas with dots
AUTHORS: OK. Changed
L205 p>0.05
AUTHORS: OK. Changed
L217 47 bp
AUTHORS: OK. Changed
I suggest to move Figure S1 to the main manuscript text, since it is important data. Also divide Figure to (a) and (b) parts, not left and right
AUTHORS: OK. Done. We completely agree and also another referee suggested the same change. So, Figure S1 is now Figure 3 in the revised version of the manuscript.
L255-258 also the single red circle is in the upper part of the network, this should be mentioned.
AUTHORS: Done.
CONCLUSIONS part is too long, please shorten it, some sentences such as L147-148 might me excluded.
AUTHORS: Ok, it has been modified. We understood that you mean L347-348 and it has been erased. Several sentences were erased too, and now the Conclusion section is shorter.
Reviewer 3 Report
Comments and Suggestions for Authors
The manuscript 'Mitochondrial Variation of Bottlenose Dolphins (Tursiops truncatus) from the Canary Islands Suggests a Key Population for Conservation with High Connectivity Within the North-East Atlantic Ocean' by Gómez-Lobo et al. is a genetic study relating to the Tursiops truncatus population of the Canary Islands. The work, although of moderate interest and with overall sufficiently predictable results, is well executed and carried out impeccably and could nevertheless be accepted on Animals but requires some small adjustments relating to the figures. The results obtained, in relation to the importance in the conservation strategies of the species, are probably a little overestimated, but still worthy of note.
specific changes to be made:
Keywords: Insert 'Tursiops truncatus' and remove 'Cetaceans'
Fig. 1: The figure must be redesigned including another insertion of the same in a broader context, the Atlantic Ocean, Europe, Africa, etc, and including the compass
Line 197: correct Canaray as Canary
Fig. 2: Enlarge the square with the captions of Fig. 2B, in the same character size as the square of Fig. 2A
Lines 224, 291: If you cite Fig. S1 in the text it must be included in that, not just in the supplementary material
Author Response
R3: Comments and Suggestions for Authors
The manuscript 'Mitochondrial Variation of Bottlenose Dolphins (Tursiops truncatus) from the Canary Islands Suggests a Key Population for Conservation with High Connectivity Within the North-East Atlantic Ocean' by Gómez-Lobo et al. is a genetic study relating to the Tursiops truncatus population of the Canary Islands. The work, although of moderate interest and with overall sufficiently predictable results, is well executed and carried out impeccably and could nevertheless be accepted on Animals but requires some small adjustments relating to the figures. The results obtained, in relation to the importance in the conservation strategies of the species, are probably a little overestimated, but still worthy of note.
AUTHORS: Thank you very much for all your comments and suggestions. We made all of them and we hope the manuscript is now suitable for publication.
Specific changes to be made:
Keywords: Insert 'Tursiops truncatus' and remove 'Cetaceans'
AUTHORS: Done.
Fig. 1: The figure must be redesigned including another insertion of the same in a broader context, the Atlantic Ocean, Europe, Africa, etc, and including the compass
AUTHORS: Thank you for your comment. The figure was modified accordingly.
Line 197: correct Canaray as Canary
AUTHORS: Done.
Fig. 2: Enlarge the square with the captions of Fig. 2B, in the same character size as the square of Fig. 2A
AUTHORS: Thank you for your comment. The figure was modified accordingly.
Lines 224, 291: If you cite Fig. S1 in the text it must be included in that, not just in the supplementary material
AUTHORS: OK. It is now included in the manuscript.
Round 2
Reviewer 1 Report
Comments and Suggestions for Authors
I think the paper has improved in quality and clarity.